# Auxin-regulated chromatin switch directs acquisition of flower primordium founder fate

**Miin-Feng Wu[1†], Nobutoshi Yamaguchi[1†‡], Jun Xiao[1†], Bastiaan Bargmann[2§], Mark Estelle[2], Yi Sang[1¶], Doris Wagner[1\*]**

[1]Department of Biology, University of Pennsylvania, Philadelphia, United States; [2]Section of Cell and Developmental Biology, Howard Hughes Medical Institute, University of California, San Diego, San Diego, United States

**\*For correspondence:** wagnerdo@sas.upenn.edu

[†]These authors contributed equally to this work

**Present address:** [‡]Graduate School of Biological Sciences, Nara Institute of Science and Technology, Ikoma, Japan; [§]Cibus US LLC, San Diego, United States; [¶]Key Laboratory of Cell Activities and Stress Adaptations, School of Life Sciences Lanzhou University, Lanzho, China

**Competing interests:** The authors declare that no competing interests exist.

**Abstract** Reprogramming of cell identities during development frequently requires changes in the chromatin state that need to be restricted to the correct cell populations. Here we identify an auxin hormone-regulated chromatin state switch that directs reprogramming from transit amplifying to primordium founder cell fate in *Arabidopsis* inflorescences. Upon auxin sensing, the MONOPTEROS transcription factor recruits SWI/SNF chromatin remodeling ATPases to increase accessibility of the DNA for induction of key regulators of flower primordium initiation. In the absence of the hormonal cue, auxin sensitive Aux/IAA proteins bound to MONOPTEROS block recruitment of the SWI/SNF chromatin remodeling ATPases in addition to recruiting a co-repressor/ histone deacetylase complex. This simple and elegant hormone-mediated chromatin state switch is ideally suited for iterative flower primordium initiation and orchestrates additional auxin-regulated cell fate transitions. Our findings establish a new paradigm for nuclear response to auxin. They also provide an explanation for how this small molecule can direct diverse plant responses.

## Introduction

Flowers are important for plant reproductive success and for human sustenance. Primordia that give rise to flowers initiate from the organogenic region of the shoot apex that surrounds the central stem cell pool (*Smyth et al., 1990*). Flower primordium initiation requires a switch from stem cell descendent (transit amplifying cell) to primordium founder cell fate (*Barton, 2010*). Primordium founder fate is promoted by a local maximum of the hormone auxin and by the AUXIN RESPONSE FACTOR (ARF) MONOPTEROS (MP/ARF5) (*Przemeck et al., 1996*). In the absence of auxin or MP, shoot apices cannot initiate flower primordia and give rise to characteristic 'naked pin' inflorescences (*Okada et al., 1991*; *Przemeck et al., 1996*; *Vernoux et al., 2000*; *Cheng et al., 2006*). Recently, targets of MP have been identified that promote flower initiation; these include a central regulator of floral fate, *LEAFY* (*LFY*), and two regulators of flower growth, *AINTEGUMENTA* (*ANT*) and *AINTE-GUMENTA-LIKE 6* (*AIL6*) (*Cole et al., 2009*; *Zhao et al., 2010*; *Yamaguchi et al., 2013*; *Besnard et al., 2014*; *Furutani et al., 2014*).

   Aux/IAA proteins together with co-repressors and repressive chromatin regulators prevent unlicensed auxin response gene expression. In the absence of the auxin stimulus, Aux/IAA proteins associate with the C-terminal domain of MP bound at its target loci (*Tiwari et al., 2003*; *Guilfoyle and Hagen, 2012*; *Yamaguchi et al., 2013*). Aux/IAA proteins directly recruit the transcriptional co-repressor TOPLESS (TPL), which in turn interacts with the histone deacetylase HDA19 (*Long et al., 2006*; *Szemenyei et al., 2008*). Histone deacetylation promotes a tight association between histones and the DNA, thus generating a chromatin state refractory to transcription (*Eberharter and*

**eLife digest** Plants form new structures such as flowers or branches throughout their life as they develop and grow. However, most plant cells are not able to produce a new flower or branch because the genes involved in these processes are usually switched off. The genes are found in regions of chromatin—the structure in which DNA is packaged in plant cells—that are normally tightly packed. This packing prevents other proteins called transcription factors from accessing the DNA and switching the genes on.

New flowers form from cells that contain high levels of a plant hormone called auxin. In these cells, a protein called MONOPTEROS switches on genes involved in making flowers. How the structure of the chromatin that surrounds these genes is altered so that they can be switched on is not clear. Wu, Yamaguchi, Xiao et al. studied this question in a plant known as *Arabidopsis*.

The experiments show that MONOPTEROS plays a crucial role in altering the structure of chromatin to allow flowers to form. In the presence of high levels of auxin, MONOPTEROS recruits groups of proteins called SWI/SNF remodeling complexes to regions of chromatin that contain genes involved in flower formation. These protein complexes loosen the structure of the chromatin so that genes can be switched on by transcription factors.

Wu, Yamaguchi, Xiao et al.'s findings suggest that auxin, with the help of MONOPTEROS and the SWI/SNF remodeling complexes, enables flower formation by changing the chromatin state. They further suggest that this chromatin state switch is also involved in leaf formation and other processes in plants that are controlled by MONOPTEROS and auxin.

*Becker, 2002*). Upon auxin sensing, Aux/IAA proteins are rapidly degraded via the SCF[TIR1/AFB] ubiquitin ligase, whose substrate recognition F-box module TIR1/AFB binds Aux/IAA proteins in the presence of auxin (*Gray et al., 2001*; *Ramos et al., 2001*; *Salehin et al., 2015*). Aux/IAA degradation leads to dissociation of the co-repressor and HDA19; this is thought to free MP to activate gene expression (*Chapman and Estelle, 2009*). How MP can execute this important function in the context of the repressive chromatin environment generated by HDA19 is not understood.

Here we uncover a new paradigm for auxin-directed transcriptional and cell fate reprogramming. The reprogramming from transit amplifying to primordium founder cell fate depends on MP-anchored chromatin unlocking by SWI/SNF ATPases. This allows additional transcription factors access to cis regulatory elements previously occluded by nucleosomes. Genetic experiments indicate that SWI/SNF recruitment is an essential function of MP and that SWI/SNF ATPase activity is necessary for reprogramming. Unlicensed chromatin remodeling at MP target loci is prevented by auxin sensitive Aux/IAA proteins, which physically block chromatin remodeler recruitment when complexed with MP. We provide evidence that that the uncovered mechanism underlies additional auxin-controlled cell fate reprogramming events, during embryos patterning and leaf morphogenesis for example.

## Results

### SWI/SNF ATPases activity is essential for flower primordium initiation

To identify factors that enable auxin-dependent activation of gene expression by overcoming the repressive chromatin at MP target loci, we screened mutants in chromatin regulators for defects in flower primordium initiation. Double hypomorph (*brm-3 syd-6*) and hypomorph/null (*brm-3 syd-5*) mutants in two related *Arabidopsis* SWI/SNF subgroup ATPases *BRAHMA* (*BRM*) and *SPLAYED* (*SYD*) formed inflorescence 'pins' characteristic of auxin pathway mutants (*Figure 1A,B*). *BRM* and *SYD* are both expressed in incipient flower primordia in the inflorescence (*Wagner and Meyerowitz, 2002*; *Wu et al., 2012*). *brm-1 syd-5* double null mutants are embryonic lethal (*Bezhani et al., 2007*). To be able to assess the flower primordium initiation in plants that have lost most SYD and BRM activity, we employed the *syd-5* null mutant and a conditional *BRM* mutant, generated by expressing an artificial micro RNA (aMIR) against BRM in adult plants (*Wu et al., 2012*). aMIRBRM reduces *BRM* expression in incipient flower primordia (*Wu et al., 2012*). *syd-5* aMIRBRM plants

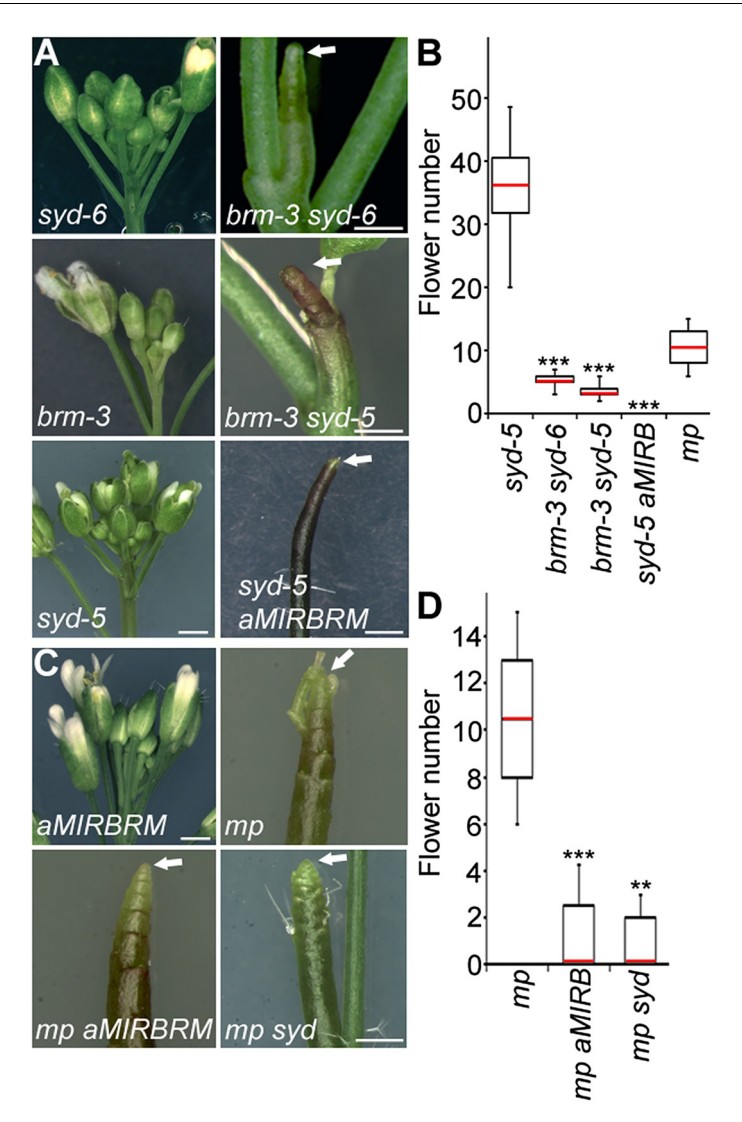

**Figure 1.** An essential role for SWI/SNF ATPases in flower primordium initiation. (**A**) 'Pin' inflorescence phenotype (white arrow) of *brm syd* double mutants. Scale bars = 1 mm. Allelic strength of mutants: *syd-6,* very weak; *brm-3* weak; *aMIRBRM,* strong; *syd-5* null. (**B**) Quantification of the flower primordia initiated in (**A**). n > 18. p-value: Mann–Whitney *U* test. (**C**) Enhancer tests using hypomorph *mp-S319* mutant (*Schlereth et al., 2010*). White arrows point to 'pin' inflorescences. Scale bars = 1 mm. *brm-1* null mutant (*Hurtado et al., 2006*) combined with the *mp-S319* hypomorph mutant is seedling lethal like the *mp-B4149* null mutant (*Weijers et al., 2006*) (*Figure 1—figure supplement 1*) and has developmental defects in the embryo (*Figure 1—figure supplement 2*). (**D**) Quantification of the flower primordia initiated in (**C**). n > 5. p-value: Mann–Whitney *U* test.

The following figure supplements are available for figure 1:

**Figure supplement 1.** *brm-1 null mutant* mutants enhance *mp-S319* hypomorph seedling phenotypes to phenocopy *mp-B4149* null mutants.

**Figure supplement 2.** *brm-1* null mutant enhance *mp-S319* hypomorph embryo phenotypes.

displayed a very dramatic flowerless 'pin' phenotype (*Figure 1A,B*). We next tested whether loss of either BRM or SYD function, neither of which causes a flower primordium initiation defect on its own

(*Figure 1A,C*), enhance the flower initiation defect of the hypomorph *mp-S319* allele (*Schlereth et al., 2010*). Hypomorph mutant phenotypes can be enhanced by loss-of-function in factors that act the same pathway. *syd-5* significantly enhanced the primordium initiation defect of the *mp-S319* mutant (*Figure 1C,D*). We could not assess flower primordium initiation in double mutants between the *brm-1* null allele and *mp-S319* because these plants phenocopied the seedling lethality of the *mp-B4149* null mutant (*Figure 1—figure supplement 1*) (*Weijers et al., 2006*). However, loss of *BRM* function in adult plants (aMIR*BRM*) significantly enhanced the defect in the flower primordium initiation of *mp-S319* (*Figure 1C,D*). The combined data indicate that SWI/SNF ATPase activity is essential for flower primordium initiation.

## BRM and SYD bind to critical MP targets and are required for their activation

One possible explanation for the striking pin inflorescence phenotype of *brm-3 syd-5* double mutants could be that BRM/SYD are required for *MP* mRNA accumulation in the organogenic region. In situ hybridization did not reveal a visible reduction of *MP* expression in *brm-3 syd-5* shoot apices (*Figure 2—figure supplement 1*). Alternatively, BRM/SYD may enable MP to activate its target genes. If this were the case, *brm-3 syd-5* and *mp-S319* should have similar molecular phenotypes. Indeed, expression of the known MP targets *LFY* and *ANT* was similarly reduced in *mp-S319* and *brm-3 syd-5* mutants (*Figure 2A*). Prior studies suggested that additional MP targets with a role in flower primordium initiation remain unidentified (*Yamaguchi et al., 2013*). We therefore tested expression of two candidate regulators of flower primordium initiation in *mp-S319* and *brm-3 syd-5*. *TARGET OF MONOPTEROS 3* (*TMO3*) is a direct target of MP during embryo development (*Schlereth et al., 2010*) that we found to be expressed in the organogenic region of the reproductive shoot apex (*Figure 2B*). *FILAMENTOUS FLOWERS* (*FIL*) encodes a regulator of organ polarity, whose expression changes dramatically during flower initiation (*Heisler et al., 2005*). Expression of both genes was strongly reduced in *mp-S319* and *brm-3 syd-5* mutants (*Figure 2A*) in further support of the idea that BRM/SYD may enable MP target gene activation. The gene expression defects were apparent in the organogenic region of shoot apices just prior to the manifestation of the morphological defect (*Figure 2B*). To further examine the role of MP in regulation of *LFY*, *ANT*, *FIL* and *TMO3* expression, we tested the effect of a steroid inducible gain or loss of MP activity in inflorescences (*Yamaguchi et al., 2013*). *LFY*, *ANT*, *FIL* and *TMO3* accumulation increased shortly after elevating and decreased shortly after reducing MP activity (*Figure 2—figure supplement 2*).

On the basis of chromatin immunoprecipitation (ChIP), MP binds to the *LFY* and *ANT* loci in inflorescences (*Yamaguchi et al., 2013*) and to the *TMO3* locus in seedlings (*Schlereth et al., 2010*). We performed MP ChIP to test whether the *TMO3* locus was bound in inflorescences and whether MP also associates with the regulatory region of the *FIL* locus. MP bound both loci in inflorescences (*Figure 2C*, *Figure 2—figure supplement 3*, *4*). Thus, *LFY*, *FIL*, *TMO3* and *ANT* are directly regulated by MP. We next tested, using ChIP, whether BRM and SYD occupy the regulatory regions of these MP target loci. BRM and SYD associated strongly with the *LFY*, *FIL*, *TMO3* and *ANT* loci (*Figure 2C*; *Figure 2—figure supplement 3*). Finally, we monitored the occupancy of MP, BRM and SYD at different sites throughout the *FIL*, *TMO3* and *LFY* regulatory regions by ChIP. MP, BRM and SYD exhibited a similar binding pattern at all loci tested (*Figure 2—figure supplement 4*). We conclude that BRM/SYD and MP occupy similar sites at shared target loci and are required for their transcriptional activation.

## FIL contributes to flower primordium initiation

Because *FIL* expression was dramatically reduced in both *mp-S319* and *brm-3 syd-5* mutants, we next wished to test whether FIL plays a role in flower initiation. *fil-8* null mutants (*Goldshmidt et al., 2008*) significantly enhanced the *mp-S319* hypomorph mutant flower initiation defect (*Figure 3A,B*). We reasoned the *syd-5* null mutants, which show no flower initiation defect on their own due to the redundant role of BRM (*Figure 1*), should also be enhanced by loss of *FIL* activity. Indeed, *syd fil* mutants formed significantly fewer flowers than the parental lines (*Figure 3C,D*). Higher order mutants in MP targets, such as *lfy ant ail-6*, form pin inflorescences when treated with a low dose of the auxin transport inhibitor NPA (*Yamaguchi et al., 2013*). Likewise, *lfy fil* double mutants formed

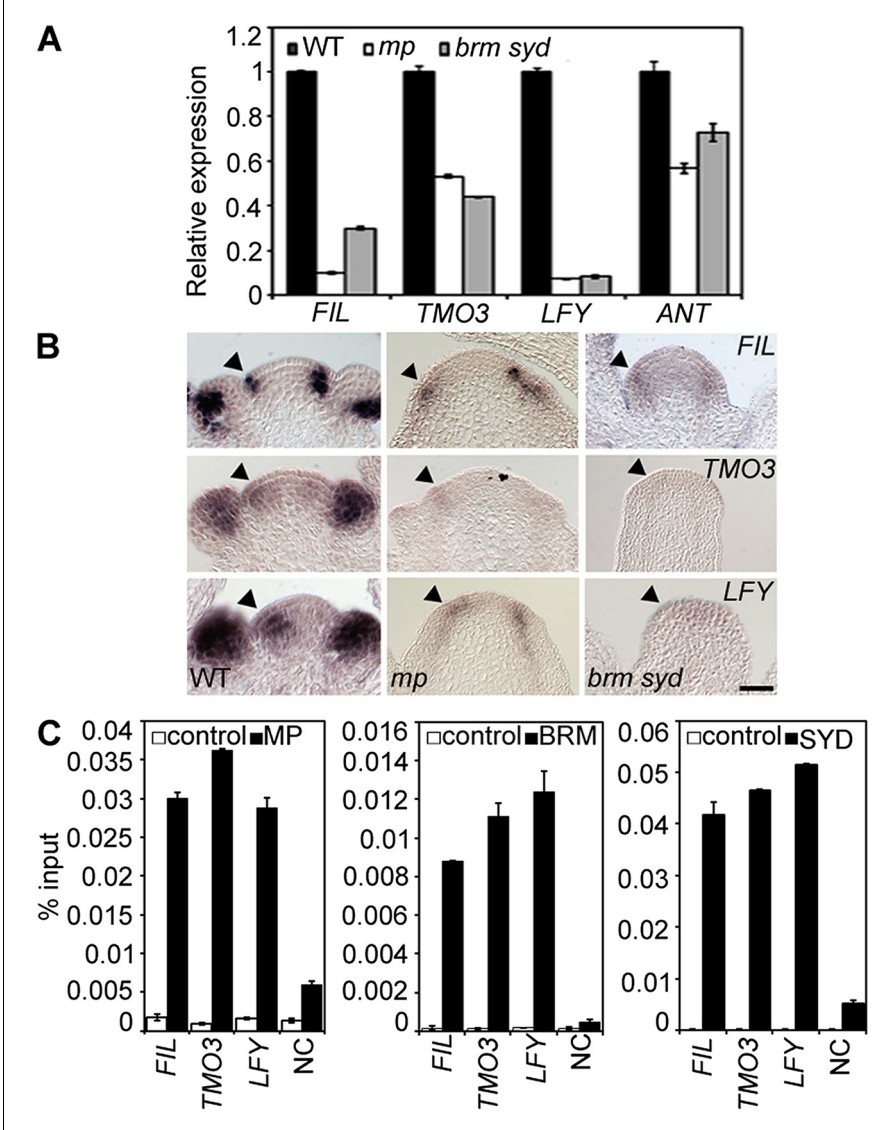

**Figure 2.** MP and BRM/SYD directly regulate common target genes. (**A**) Expression levels of *FIL, TMO3, LFY* and *ANT* in wild-type (WT), *mp-S319* or *brm-3 syd-5* inflorescence shoot apices normalized to that of *EIF4A-1*. Expression in WT was set to one. (**B**) In situ hybridization of wild-type, *mp-S319* or *brm-3 syd-5* inflorescence shoot apices prior to 'pin' formation using antisense *FIL, TMO3* and *LFY* probes. Black arrowheads: organogenic region from where flower primordia usually arise. *Figure 2—figure supplement 1* shows that *MP* expression is not visibly reduced in *brm-3 syd-5* mutants. Inducible increase or reduction of MP function triggered increased or decreased expression of *FIL, TMO3, LFY* and *ANT*, respectively (*Figure 2—figure supplement 2*). (**C**) Anti-GFP chromatin immunoprecipitation (ChIP) to test pSYD:GFP-SYD and pBRM:BRM-GFP occupancy at pMP:MP-HA bound sites (as determined by anti-HA ChIP). For MP, BRM and SYD occupancy at the *ANT* locus see *Figure 2—figure supplement 3*. For comparison of the binding pattern of BRM, SYD and MP at the *FIL, TMO3* or *LFY* loci see *Figure 2—figure supplement 4*. Control: anti-GFP or anti-HA ChIP in non-transgenic plants. NC: negative control locus (*Ta3* retrotransposon).

The following figure supplements are available for figure 2:

**Figure supplement 1.** *MP* expression in wild type and *brm-3 syd-5* mutant inflorescences.

**Figure supplement 2.** Elevated MP activity leads to increased and reduced MP activity to decreased accumulation of *LFY, TMO3, FIL* and *ANT*.

*Figure 2 continued on next page*

*Figure 2 continued*

**Figure supplement 3.** BRM and SYD occupancy at known MP target loci.

**Figure supplement 4.** BRM, SYD, or MP occupancy at different regions of the *FIL, TMO3* and *LFY* loci.

inflorescence pins when treated with a low dose of NPA (*Figure 3E,F*). Thus, the direct MP target *FIL* contributes to initiation of flower primordia.

## MP physically interacts with and may recruit BRM/SYD

BRM and SYD each are the catalytic subunit of a multiprotein chromatin remodeling complex (reviewed in *Han et al., 2015*). To test whether MP recruits chromatin remodeling complexes formed around BRM or SYD to its target loci to overcome the repressed chromatin state, we examined whether MP physically interacts with either chromatin remodeling complex. Bimolecular fluorescence complementation (BiFC) and co-immunoprecipitation (co-IP) revealed that MP interacts with the BRM- and the SYD-containing complex (*Figure 4A,B*) in plant cells. The interaction was enhanced by auxin application (*Figure 4B,C*; *Figure 4—figure supplement 1*). No BiFC signal was observed when we used a version of MP that consisted solely of the N-terminal domain (*Figure 4A, C*; *Figure 4—figure supplement 2*). Yeast-two-hybrid tests with MP and BRM revealed that no other plant proteins are required for the physical interaction and allowed us to map the interacting region of MP to its middle domain (*Figure 4D*), which is critical for transcriptional activation (*Tiwari et al., 2003*). We used the in situ proximity ligation assay (PLA), an immunoassay that allows visualization of protein interactions in tissue sections (*Soderberg et al., 2008*), to examine where at the shoot apex MP interacts with BRM. On the basis of in situ PLA, MP associates with BRM specifically in the organogenic region of the shoot apex from where flower primordia initiate (*Figure 4E*). No signal was detected when we performed the PLA assay in plants only expressing pBRM:BRM-GFP (*Figure 4E*). To directly test whether MP activity is required for BRM and SYD binding to its target loci, we employed ChIP in wild-type and *mp-S319* mutant inflorescences. In vivo association of BRM or SYD with the *FIL* and *LFY* loci was much reduced in *mp-S319* inflorescences (*Figure 4F*). The data are consistent with the hypothesis that MP may recruit BRM/SYD to target loci.

## SWI/SNF ATPases 'unlock' the chromatin for transcriptional activation and flower primordium initiation

Studies in embryos had suggested that MP-interacting Aux/AA proteins recruit the transcriptional co-repressor TPL and the histone deacetylase HDA19 to MP target loci to prevent MP from activating its target genes when auxin levels are low (*Long et al., 2006*; *Szemenyei et al., 2008*). We found that TPL and HDA19 occupied the MP-bound sites at the *LFY* and *FIL* loci in inflorescence apices in the absence but not in the presence of auxin application, as expected (*Figure 5—figure supplement 1A,B*). In addition, auxin treatment led to increased histone 3 lysine (acetylation [H3K9ac], an activating histone modification removed by HDA19 [*Krogan et al., 2012*], at both loci [*Figure 5—figure supplement 1C*]).

We next tested whether BRM and SYD are required for overcoming the repressed chromatin state generated by TPL and HDA19. BRM or SYD belong to the SWI/SNF subgroup chromatin remodelers, which alter accessibility of the genomic DNA by changing the occupancy or positioning of nucleosomes (*Clapier and Cairns, 2009*; *Han et al., 2012*). To assess the accessibility of the MP bound regions at the *FIL* and *LFY* loci in inflorescences, we employed Formaldehyde Assisted Isolation of Regulatory Elements (FAIRE), a method that enriches accessible (nucleosome depleted) genomic DNA from crosslinked chromatin after phenol/chloroform extraction (*Simon et al., 2012*). FAIRE revealed increased accessibility at the *FIL* and *LFY* loci after exogenous auxin application (*Figure 5A*). Likewise, auxin treatment triggered increased H3K9 acetylation at both loci and caused increased *FIL* and *LFY* mRNA accumulation (*Figure 5B,C*). Auxin treatment failed to increase *FIL* and *LFY* locus accessibility, presence of activating histone marks, and gene expression in *brm-3 syd-5* mutant inflorescences (*Figure 5A–C*). The combined data suggest that BRM or SYD are necessary

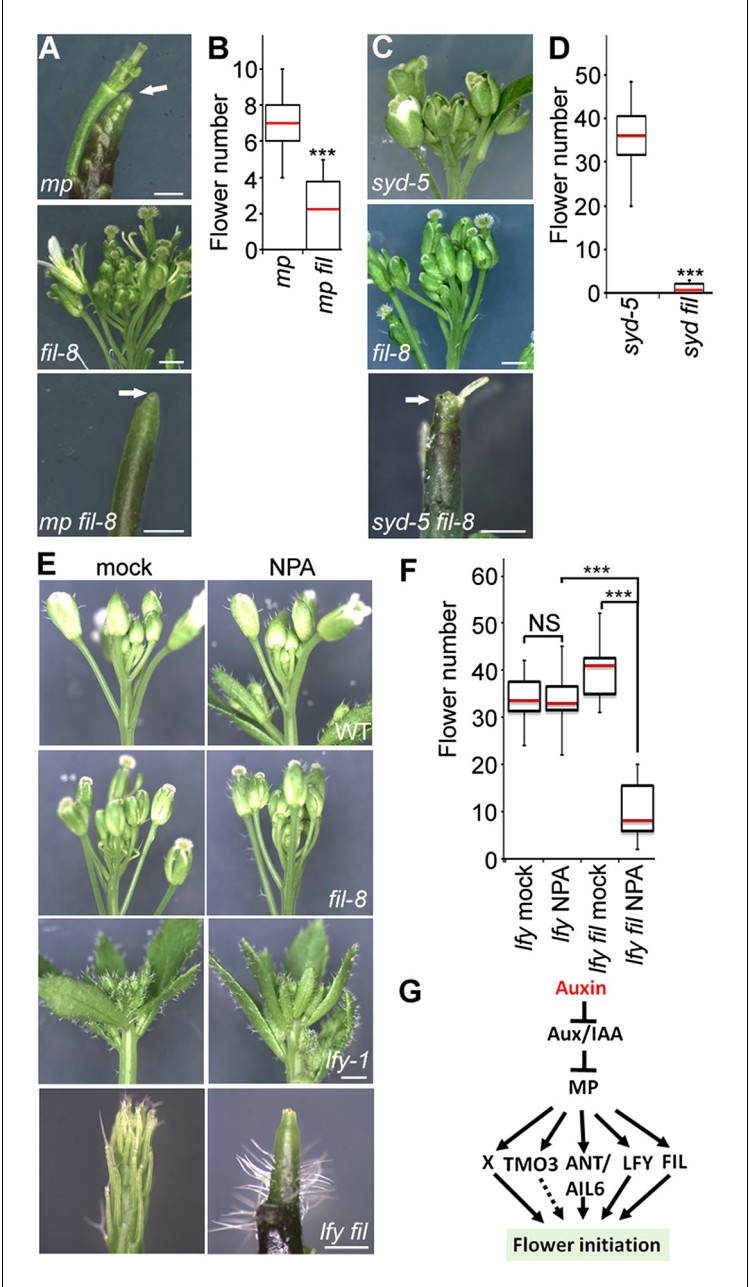

**Figure 3.** The direct MP and BRM/SYD target *FIL* plays a role in flower primordium initiation. (**A**) Enhancer test using the hypomorph *mp-S319* and the null *fil-8* mutant. Scale bars = 1 mm. White arrows point to pin inflorescences. (**B**) Quantification of flower primordia initiated in (**A**). n > 10. p-value: Mann–Whitney *U* test. (**C**) Enhancer test using null *syd-5* mutant and *fil-8*. White arrow points to pin-like inflorescence. Scale bars = 1 mm. (**D**) Quantification of flower primordia initiated in (**C**). n > 5. p-value: Mann–Whitney *U* test. (**E**) 'Pin' inflorescence phenotype of *lfy-1 fil-8* double mutant treated with the auxin transport inhibitor N-1-naphthylphthalamic acid (NPA). Scale bars = 1 mm. (**F**) Quantification of the flower primordia initiated in (**E**). n > 12. p-value: Mann–Whitney *U* test. (**G**) Updated model for auxin/MP-mediated flower primordium initiation together with BRM/SYD. Dashed arrow: role not yet proven. X: additional MP target(s) with a role in flower initiation.

for the auxin-dependent increase in accessibility at MP target loci in the context of chromatin, a prerequisite for induction of MP targets.

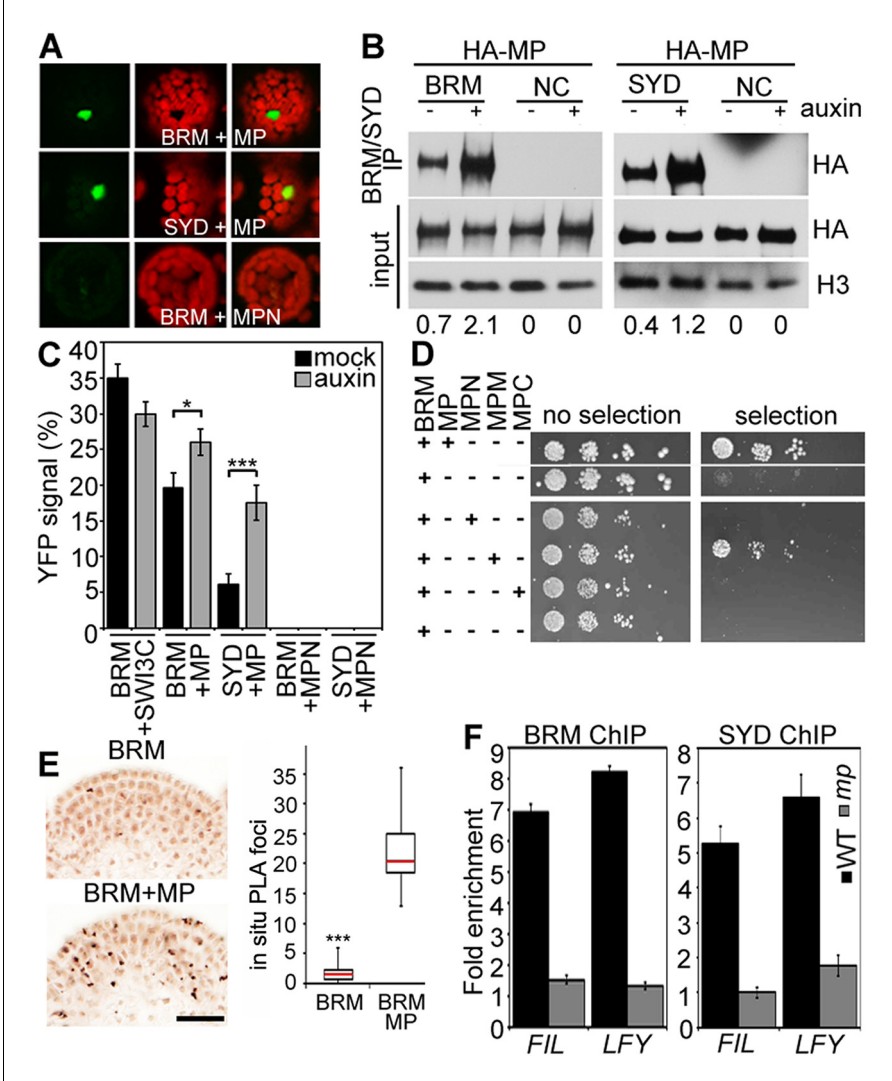

**Figure 4.** MP physically interacts with and recruits BRM and SYD to target loci. (A) Bimolecular fluorescence complementation (BiFC) test of MP and BRM or SYD protein interaction in plant cells. Green: BiFC signal in the nucleus, red: chloroplast auto-fluorescence. MPN: N-terminal domain of MP. (B) Co-immunoprecipitation using anti-FLAG antibody in plant cells expressing HA-MP with or without FLAG-BRM or FLAG-SYD. Western blot is probed with anti-HA or anti-histone H3 antibody. Below: Amount of precipitated HA-MP (% of input). See also *Figure 4—figure supplement 1*. (C) Quantification of BiFC events in the absence or presence of auxin. The error bars are proportional to the standard error of the pooled percentage computed using binomial distribution. n = 3. p-value; Mann–Whitney *U* test. SWI3C: BRM chromatin remodeling complex component (positive control). (D) Yeast-two-hybrid test of interaction between BRM and MP or MP domains: N: N-terminus, M: middle region, C: C-terminus. See *Figure 4—figure supplement 2* for domains of the MP protein. Growth was assayed minus (left) or plus (right) 3-amino-1,2,4-triazole. Thin white line: cropped image from one plate. (E) In situ proximity ligation assay (PLA) with anti-GFP and anti-HA antibodies in pBRM:BRM-GFP or pMP:MP-HA pBRM:BRM-GFP shoot apices. Left: individual sections, right: quantification of interaction foci. n > 12. p-value: Student's *t*-test. (F) BRM and SYD ChIP enrichment at the *FIL* and *LFY* loci relative to the control locus (*Ta3* retrotransposon) in wild-type and *mp-S319* mutant inflorescences.

The following figure supplements are available for figure 4:

**Figure supplement 1.** Auxin treatment enhanced the physical interaction between BRM and MP.

**Figure supplement 2.** Domains of MONOPTEROS.

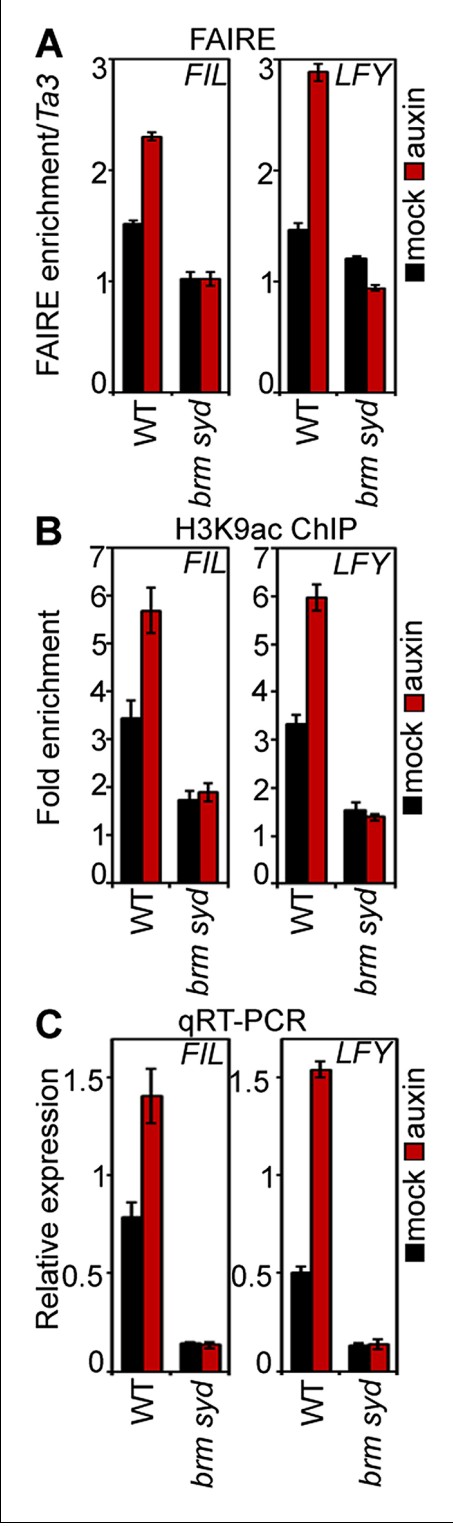

**Figure 5.** SWI/SNF chromatin remodeling ATPases are required for increased accessibility of MP target loci in response to auxin. (**A**) DNA accessibility at *FIL* and *LFY* loci in the context of chromatin assayed by Formaldehyde Assisted Isolation of Regulatory Elements (FAIRE) in response to auxin treatment in wild-type (WT) and *brm-3 syd-5* inflorescences. The *Figure 5 continued on next page*

## SWI/SNF ATPases tethering causes increased accessibility of target loci, transcriptional activation and flower primordium initiation

To test whether BRM/SYD recruitment leads to induction of MP target genes, we tethered the chromatin remodeling complexes to MP targets by fusing the N-terminal DNA binding domain of MP (*Tiwari et al., 2003*; *Boer et al., 2014*) to a shared component of the BRM and the SYD chromatin remodeling complex (*Han et al., 2015*) called BUSHY (MPN-BSH; *Figure 6A*). MPN-BSH transfection into plant cells caused an increase in endogenous *FIL* expression (*Figure 6B*). The magnitude of the response was comparable to that observed upon auxin application (*Figure 6B*). *FIL* mRNA levels did not increase when we transfected MPN alone or when we transfected MPN-BSH into *brm-3 syd-5* cells, suggesting that the observed activity of the MPN-BSH fusion protein depends on its ability to recruit BRM/SYD (*Figure 6B*; *Figure 6—figure supplement 1*). MPN-BSH activity apparently did not require interaction with endogenous MP because introducing a mutation that interferes with homodimerization (*Boer et al., 2014*) (MPNm1-BSH) did not impair activity (*Figure 6—figure supplement 1*). By contrast, introducing a second mutation, which abolishes DNA binding specificity, (*Boer et al., 2014*) (MPNm2-BSH), blocked activity of the fusion protein (*Figure 6B*). The combined data indicate that BRM/SYD tethering via MPN-BSH is sufficient to induce *FIL* expression.

Next, we monitored the effect of auxin treatment and BRM/SYD tethering on accessibility of the *FIL* locus. Auxin-treatment or MPN-BSH-transfection caused increased accessibility of the endogenous *FIL* locus regulatory region on the basis of FAIRE (*Figure 6C*). We employed limited micrococcal nuclease (MNase) digestion and tiled oligo qPCR to identify a well-positioned nucleosome near the MP and BRM/SYD bound site at the *FIL* locus (*Figure 6D*). Auxin treatment or MPN-BSH transfection led to strong destabilization of this nucleosome on the basis of MNase-qPCR in plant cells (*Figure 6E*). The slightly stronger nucleosome destabilization observed upon auxin treatment was expected; while all cells can respond to auxin, only those transfected (40% on average) can respond to MPN-BSH. Auxin treatment in *brm syd* mutant cells did not lead to destabilization of the well-positioned nucleosome at the *FIL* locus (*Figure 6—figure supplement 1*). We conclude that tethering of BRM or SYD

*Figure 5 continued*

ratio of FAIRE enrichment at the locus of interest was normalized over that at the *Ta3* retrotransposon. (B) Anti-histone 3 lysine 9 acetylation (H3K9ac) ChIP at the *FIL* (left) and *LFY* (right) locus normalized over that at *Ta3* in genotypes and treatments shown in (A). (C) *FIL* and *LFY* RNA accumulation relative to *EIF4A-1* in genotypes and treatments shown in (A).
The following figure supplement is available for figure 5:

**Figure supplement 1.** TPL/HDA occupancy and H3K9ac levels at the *FIL* and *LFY* target loci with and without auxin application.

complexes to MP target loci increases their accessibility and transcription.

Finally, we tested whether BRM/SYD tethering can rescue flower primordium initiation in the hypomorph *mp-S319* mutant. MPN-BSH and the dimerization defective version MPNm1-BSH caused nearly complete rescue of the flower initiation defects of *mp-S319* (*Figure 6F,G*; *Figure 6—figure supplement 2*). By contrast, MPNm2-BSH, which has no DNA binding specificity, did not increase flower initiation in *mp-S319* mutant plants (*Figure 6F,G*). Likewise, MPN alone, which cannot recruit BRM/SYD, did not rescue the *mp-S319* phenotype (*Figure 6—figure supplement 2*). The extensive rescue of *mp-S319* by MPN-BSH suggests that MP executes its essential role in flower primordium initiation in large part by recruiting BRM and SYD to target loci to 'open up' compacted chromatin. That MPN-BSH did not direct ectopic flower initiation in *mp-S319* suggests that the auxin pre-pattern is still being correctly interpreted in *mp-S319* MPN-BSH, either through the residual MP activity present in *mp-S319* (*Schlereth et al., 2010*), or through other factors. Intriguingly, MPN-BSH and MPNm1-BSH also rescued other phenotypic defects of *mp-S319* (*Figure 6—figure supplement 3*), indicating that SWI/SNF recruitment by MP underlies additional developmental processes controlled by auxin.

## An auxin-dependent, MP-anchored, chromatin state switch

Finally, we asked how chromatin remodeler recruitment is limited to cells that have experienced an auxin maximum. Since auxin treatment enhanced MP interaction with both BRM and SYD (*Figure 4B,C*; *Figure 4—figure supplement 1*), we hypothesized that auxin-sensitive Aux/IAA proteins might block the interaction between MP and the SWI/SNF ATPases. We probed the effect of two Aux/IAA proteins known to associate with MP, (BODENLOS [BDL] and AUXIN RESISTANT 3 [AXR3]) (*Ouellet et al., 2001*; *Weijers et al., 2006*), on the MP interaction with BRM. Presence of either Aux/IAA was sufficient to prevent BRM from associating with MP in yeast (*Figure 7A,B*). Likewise, auxin-insensitive versions of BDL (bdl) and AXR3 (axr3) strongly interfered with the MP-BRM interaction in plant cells on the basis of co-IP and BiFC experiments (*Figure 7C,D*). In both yeast and plant assays, only Aux/IAA proteins complexed with MP via the MP C-terminal domain effectively blocked BRM from associating with MP (*Figure 7B–D*). Finally, increased nuclear accumulation of axr3 (after steroid activation of axr3-GR) caused BRM and SYD dissociation from the *LFY* and *FIL* loci (*Figure 7E*). Thus, Aux/IAA proteins block SWI/SNF ATPase recruitment to MP target loci in the absence of the hormonal cue.

## Discussion

### A molecular framework for acquisition of flower primordium founder fate

A classical role of auxin is initiation of flower primordia from the organogenic region of the shoot apex. Flowers are critical for plant reproductive success and human sustenance. Despite its importance, mechanistic insight into the nuclear responses that underlie auxin-mediated cell fate reprogramming is lacking. We show here that after perception of the hormonal cue, the ARF MP executes its central role (*Przemeck et al., 1996*; *Reinhardt et al., 2003*) in flower primordium initiation in large part by recruiting BRM or SYD-containing chromatin remodeling complexes to its target loci to unlock compacted chromatin. Tethering SWI/SNF complexes to MP target loci led to extensive genetic rescue of *mp-S319* mutant flower initiation defects, while loss-of-function analyses uncovered an essential role for the chromatin remodeling ATPases in induction of MP target genes and in flower primordium initiation. Unlicensed activation of MP targets in the absence of the hormonal stimulus is prevented by MP interacting Aux/IAA proteins, which noncompetitively inhibit BRM or

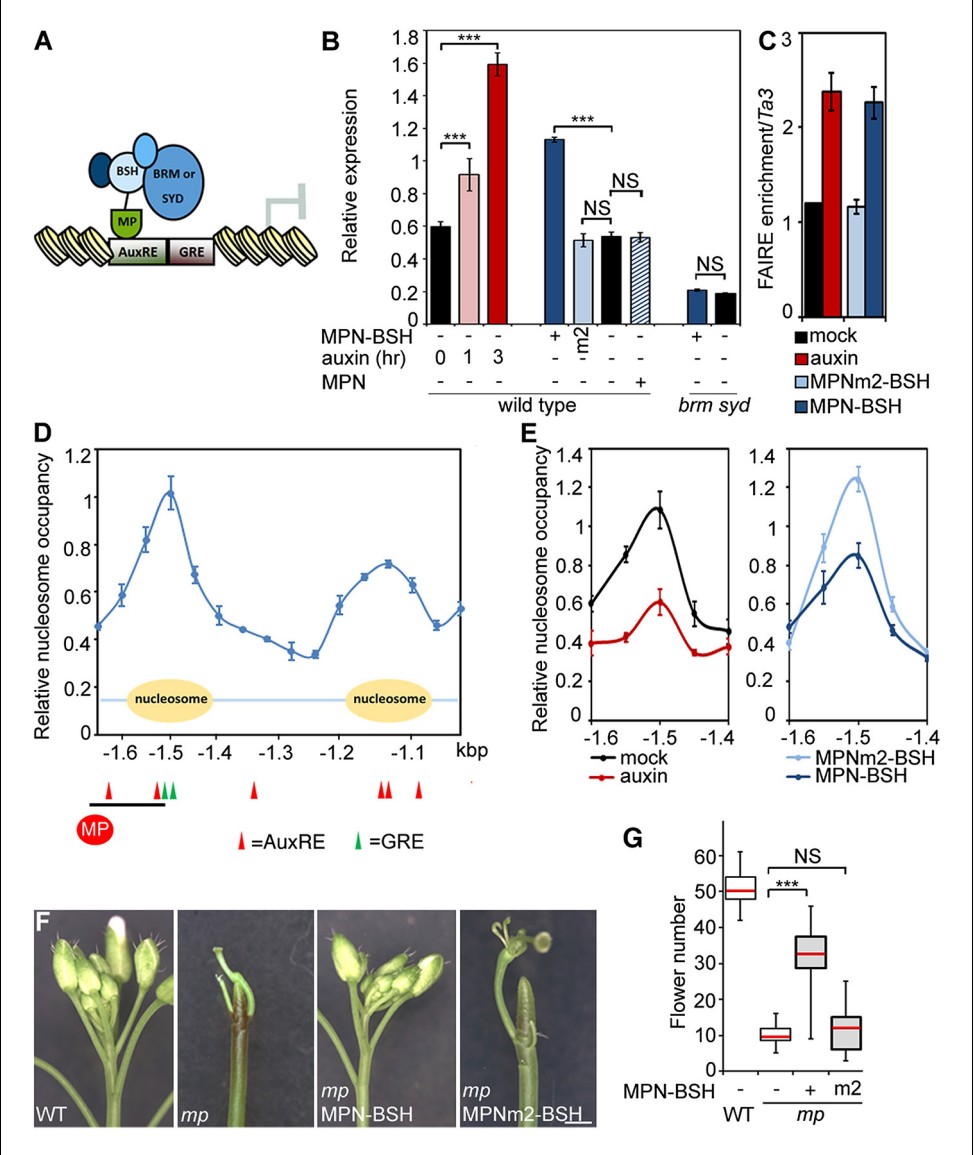

**Figure 6.** Tethering the BRM or SYD complex to MP target loci mimics MP function. (**A**) Tethering of BRM or SYD-containing SWI/SNF complexes to the MP target loci. The shared BRM and SYD complex subunit BUSHY (BSH) (**Han et al., 2015**) is translationally fused to the MP DNA binding domain (MPN-BSH). (**B**) Transcriptional activation of the *FIL* locus by auxin treatment or BRM/SYD tethering via MPN-BSH in isolated plant cells. MPNm1-BSH carries a mutation (G279E; **Figure 6—figure supplement 1**) that blocks MP dimerization (**Boer et al., 2014**). MPNm2-BSH carries a second mutation (R215A) that causes loss of DNA binding specificity. Controls: MPN, mock treatment or no plasmid. n > 3. p-value: Student's *t*-test. (**C**) DNA accessibility at the *FIL* locus in response to auxin treatment or BRM/SYD tethering assayed by FAIRE in isolated plant cells. (**D**) Nucleosome positioning at the *FIL* locus. Top: MNase digestion followed by tiled oligo qPCR (MNase-qPCR) to monitor nucleosome positioning at the *FIL* promoter in 3-week-old plants. X-axis: distance from the start codon. Middle: diagram of nucleosome positions. Bottom: red circle: MP protein. Red triangles: core MP binding sites (AuxREs) (**Ulmasov et al., 1997**; **Boer et al., 2014**). Black line: region probed in all ChIP or FAIRE experiments (*FILb* in **Figure 2—figure supplement 4**). Green triangle: Evolutionarily conserved cis elements. (**E**) Nucleosome occupancy at the *FIL* locus in response to auxin treatment or BRM/SYD tethering via MPN-BSH in isolated plant cells by MNase-qPCR. X-axis: distance from the start codon. **Figure 6—figure supplement 1** shows the nucleosome occupancy in *brm syd* mutant plant cells in response to auxin treatment. (**F, G**) Rescue of *mp-S319* flower primordium initiation defect by tethering of BRM or SYD complexes to MP binding sites. **Figure 6—figure supplement 2** shows the effect of additional rescue constructs on flower initiation in *mp-S319* mutants. **Figure 6—figure supplement 3** shows rescue of *mp-S319* mutant leaf developmental defects. (**F**) Representative inflorescence images. Scale bars = 1

*Figure 6 continued on next page*

*Figure 6 continued*
mm. (**G**) Quantification of flower primordium initiation. n > 18. Grey shading: T1 population of transgenic plants.
p-value: Mann–Whitney *U* test.
The following figure supplements are available for figure 6:

**Figure supplement 1.** Auxin treatment fails to destabilize the well-positioned nucleosome at the *FIL* locus in *syd brm* mutant plant cells.
**Figure supplement 2.** Rescue of *mp-S319* by tethering the BRM or SYD complex to MP target loci.
**Figure supplement 3.** MPN-BSH and *MPNm1-BSH* rescue *mp-S319* mutant leaf phenotypes.

SYD complex recruitment (*Figures 7* and *8*). This prevents premature overturning of the repressive chromatin state generated by the TPL/HDA19 complex (*Long et al., 2006*; *Szemenyei et al., 2008*).

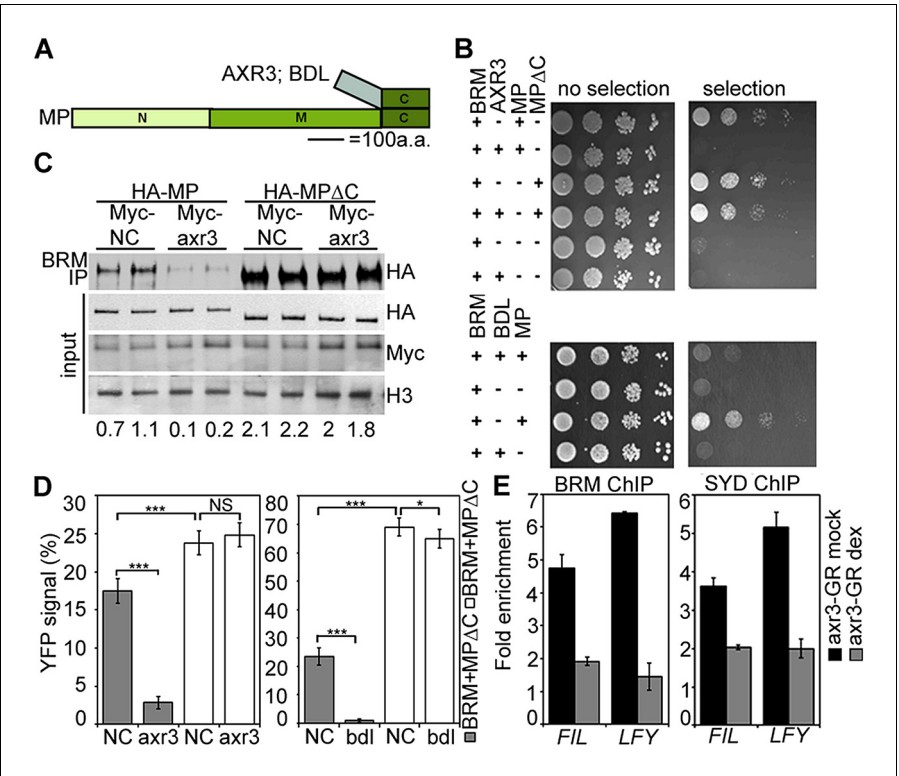

**Figure 7.** Aux/IAA proteins prevent BRM and SYD recruitment by MP. (**A**) Diagram of MP domains. N: N-terminal DNA binding/dimerization domain, M: middle BRM/SYD interacting region, C: C-terminal Aux/IAA interacting domain. (**B**) Yeast-three-hybrid test of BRM interaction with MP or MP lacking the C-terminal domain (MPΔC) in the presence of the Aux/IAA protein AXR3 (top) or BDL (bottom). Growth was assayed with (right) or without (left) 3-amino-1,2,4-triazole. (**C**) Co-immunoprecipitation of FLAG-BRM with HA-MP or HA-MPΔC in the presence of the stabilized Myc-axr3. NC: Myc-tagged unrelated protein of similar molecular mass as axr3. Below: Amount of precipitated HA-MP/HA-MPΔC (% input). (**D**) Quantification of BiFC test of interaction between BRM and MP or BRM and MPΔC in the presence of axr3 (left) or bdl (right) compared to a NC protein. The error bars are proportional to the standard error of the pooled percentage computed using binomial distribution. n = 3. p-value: Mann–Whitney *U* test. (**E**) ChIP to assess BRM and SYD association with MP target gene loci before (mock) or after (dex) nuclear entry of axr3-GR. Shown is fold-enrichment relative to a control locus (*Ta3* retrotransposon).

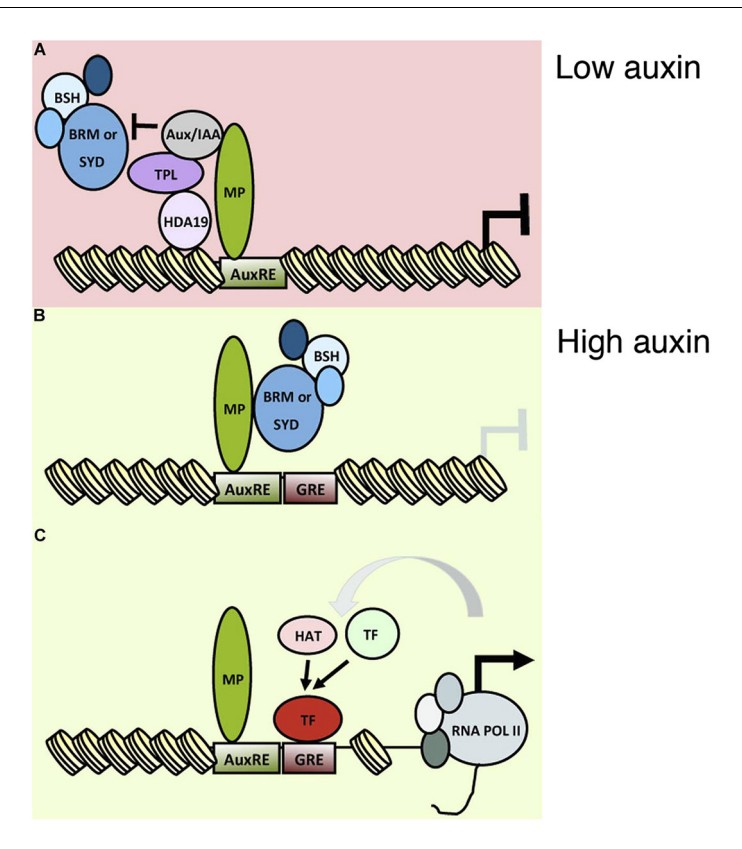

**Figure 8.** An auxin triggered chromatin state switch. (**A**) In conditions of low auxin, Aux/IAA proteins bind to MP transcription factor associated with target loci and prevent gene expression in two ways: by recruiting the co-repressor TOPLESS (TPL) and histone deacetylase HDA19 and by preventing recruitment of the BRM or SYD chromatin remodeling complexes. (**B**) Upon establishment of a local auxin maximum, Aux/IAA proteins are degraded, this leads to eviction of HDA19 and TPL. Aux/IAA degradation also frees MP to recruit BRM or SYD complexes. The chromatin remodeling complexes open up the compacted chromatin by reducing nucleosome occupancy, thus increasing the accessibility of the genomic DNA near MP bound sites. (**C**) The 'chromatin unlocking' allows additional transcription factors access to their *cis* elements. This, possibly via additional steps, leads to recruitment of the general transcriptional machinery and initiation of transcription. HAT: histone acetyl transferase. GRE: binding site for transcription factor (TF). *Figure 8—figure supplement 1* shows evolutionarily conserved cis elements near the midpoint of the well-positioned nucleosome at the *FIL* locus.

The following figure supplement is available for figure 8:

**Figure supplement 1.** Two evolutionarily conserved cis elements close to the midpoint of the well-positioned nucleosome at the *FIL* locus.

## A versatile, auxin tunable chromatin state switch for cell identity reprogramming

Our study uncovers a rapid, and precise MP-anchored chromatin state switch that underlies flower primordium initiation. These attributes make it well suited to support both iterative initiation of floral primordia as new local auxin maxima form and positioning of flower primordia at the correct phyllotactic distance from one another (*Heisler et al., 2005*; *Traas, 2013*). It is rapid because Aux/IAA proteolysis is triggered immediately upon auxin sensing (*Gray et al., 2001*; *Ramos et al., 2001*), this allows a rapid onset of the 'ON' state. In addition, BRM and SYD complexes are present in all rapidly dividing cells (*Wagner and Meyerowitz, 2002*; *Bezhani et al., 2007*; *Wu et al., 2012*). It is precise

due to the inherent auxin sensitivity of individual Aux/IAA proteins (*Calderón Villalobos et al., 2012*; *Havens et al., 2012*), this enables tuning of the switch to specific auxin thresholds.

BRM and SYD act as 'gatekeepers' for auxin-triggered transcriptional activation. Subsequent to chromatin remodeling, additional sequence specific binding proteins gain access to their previously occluded binding sites. The well-positioned nucleosome near the MP-bound site at the *FIL* locus, which is destabilized by auxin treatment or SWI/SNF tethering, is positioned over evolutionary conserved *cis* elements (*Figure 8—figure supplement 1*), some of which had previously been shown to co-occur with AuxREs (*Berendzen et al., 2012*). The transcription factors that can gain access to their binding sites only after chromatin remodeling allow additional layers of selectivity. For example, different subsets of the MP and SWI/SNF unlocked genes may be activated in different cell types based on the prevailing transcription factor repertoire. In addition, this mechanism supports a protracted response (*Mahonen et al., 2014*), if the accumulation of critical transcription factors is delayed relative to chromatin unlocking. The uncovered paradigm for auxin-triggered transcriptional activation thus helps explain how auxin can direct many different plant responses.

## A general and conserved framework for auxin controlled cell fate reprogramming

The phytohormone auxin is a key signal in plant morphogenesis, controlling most aspects of plant development and growth (*Reinhardt et al., 2000*; *Scarpella et al., 2010*; *Domagalska and Leyser, 2011*; *Salehin et al., 2015*; *ten Hove et al., 2015*). Our genetic enhancer and rescue tests implicate chromatin remodeling by BRM and SYD in embryogenesis, root development, seedling viability and leaf development (*Figure 1—figure supplement 1*; *Figure 1—figure supplement 2*; *Figure 6—figure supplement 3*). This suggests that MP-dependent recruitment of BRM/SYD and chromatin 'unlocking' is required for many developmental processes controlled by auxin and may be a general mechanism for auxin-triggered cell fate reprogramming. Other activating ARFs (*Tiwari et al., 2003*) may also recruit BRM/SYD. All components required for regulatory switch—from the SCF$^{TIR1/ABF}$ complex to ARF, Aux/IAA, BRM/SYD, TPL, HDAC—are conserved in all land plants (*Remington et al., 2004*; *De Smet et al., 2011*; *Sang et al., 2012*; *Kato et al., 2015*), suggesting the possibility that it represents an ancient regulatory module.

Reprogramming of cell identities during development frequently requires chromatin state changes (*Chen and Dent, 2014*). A key question is how general chromatin remodelers can function in a specific genomic context to change the fate of a restricted cell population in a precise, cue dependent manner (*Han et al., 2015*). Here, we uncover a simple and elegant mechanism for small-signaling-molecule-regulated chromatin state switch that is anchored to precise genomic locations by a master transcription factor, can rapidly respond to a range of signaling molecule concentrations and is versatile in that it supports diverse transcriptional and cell fate identity outcomes.

## Materials and methods

### Plant materials and treatments

Mutant alleles and transgenic plants used in this study include *brm-1* (*Hurtado et al., 2006*), *brm-3* (*Farrona et al., 2007*), *syd-5* (*Bezhani et al., 2007*), *syd-6* (*Han et al., 2012*), pLFY:aMIRBRM (*Wu et al., 2012*), *mp-S319* (*Cole et al., 2009*), *mp-B4149* (*Weijers et al., 2006*), *arf7-1* (*Okushima et al., 2005*), *ap1-1 cal-1* (*Ferrandiz et al., 2000*), *fil-8* (*Goldshmidt et al., 2008*), *lfy-1* (*Weigel et al., 1992*), pBRM:BRM-GFP and pSYD:GFP-SYD (*Wu et al., 2012*), 35S:TPL-GFP (*Long et al., 2006*). All are in the Columbia accession. Flower number was counted at 60 to 65 DAG (days after germination). For expression and ChIP, 5 cm bolt inflorescences were treated with 10 µM dexamethasone (DEX, Sigma St. Louis MO, United States) or 10 µM indole-3-acetic acid (IAA, Sigma) plus 0.015% Silwet-77. For mock treatments, 0.1% ethanol or 0.1% DMSO plus 0.015% Silwet-77 were used. Inflorescences were harvested 6 hr after treatment. For auxin treatments in protoplasts, $4 \times 10^6$ cells were harvested from leaves of 15 day-old long-day-grown plants and treated with 10 µM IAA in 0.02% ethanol in buffer W5 (*Yoo et al., 2007*). Treatment duration ranged from 15 min to 3 hr. For 1-N-Naphthylphthalamic acid (NPA, Sigma) treatments, 15-day-old seedlings were sprayed with 10 nM NPA plus 0.015% Silwet-77 or with 0.1% DMSO plus 0.015% Silwet-77 every 3 days for total of 9 treatments. To test for gene expression changes upon inducible increase or reduction in

MP activity, 14-day old long-day grown seedlings of plants expressing and estradiol inducible version of MP (pER>>MPΔC) or a dexamethasone inducible version or AXR3 (ap1cal axr3-GR) were sprayed with 10 µM β-estradiol (Sigma) in 0.05% ethanol or 10 µM dexamethasone (Sigma) in 0.05% ethanol. Mock treatment was with 0.05% ethanol. Samples were harvested at 3 hr or 6 hr after treatment.

## Transgenic plants

To generate pMP:MP-6xHA, a full-length MP genomic fragment was cloned into pENTR/D-TOPO (Thermo Fisher, Waltham MA, United States). The stop codon was replaced by a *Spe*I site and a 6xHA tag was inserted. The pMP:MP-6xHA clone was shuttled into pKGW (*Karimi et al., 2002*). To generate *axr3-GR*, the stop codon of *axr3* in pKGW was replaced by an *Nde*I site and the rat glucocorticoid receptor was inserted at the 3′ end of *axr3*. The *35S* promoter was cloned into pKGW to obtain *35S:axr3-GR* by LR clonase. To generate estradiol inducible MP, a truncated version of MP was missing the C-terminal PB1 domain (amino acids 795–902) was amplified from cDNA, cloned into pENTR/D-TOPO (Thermo Fisher) and sequence verified. The clone was shuttled into the estradiol-inducible expression vector pMDC7 (*Curtis and Grossniklaus, 2003*). To generate HDA19-GFP, a HDA19 genomic fragment was amplified and cloned into pENTR/D-TOPO. The stop codon was replaced by a *Spe*I site and the green fluorescence protein (GFP) coding region was inserted. The pHDA19:HDA19-GFP fragment was cloned into the *Not*I site of the binary vector pMLBART. To generate 35S:MPN-BSH, 35S:MPNm1-BSH and 35S:MPNm2-BSH, the N-terminal MP DNA binding domain (MPN, [amino acids 1–348]), a dimerization mutant (MPNm1, G279E, [*Boer et al., 2014*]), or a dimerization and DNA binding mutant (MPNm2, G279E R215A, [*Boer et al., 2014*]), were fused in-frame with the full length coding region of BSH (*Bezhani et al., 2007*) and sub-cloned into pUC19. The resulting MPN-BSH cDNAs were cloned into pGWB2 (*Nakagawa et al., 2007*). For 35S: MPN and 35S:MPN-BSH, the MPN and MPNm1 fragments were cloned into pENTR/D-TOPO and recombined into pGWB2 (*Nakagawa et al., 2007*). All constructs were transformed into *mp-S319/+* plants by floral dip. For primer sequences see *Supplemental file 1*.

## Expression analysis

qRT-PCR was performed as previously described (*Yamaguchi et al., 2013*). Expression levels were determined by real-time PCR and normalized over that of *EUKARYOTIC TRANSLATION INITIATION FACTOR 4A-1* (*EIF4A-1*; At3g13920). The mean and standard error were determined using three technical replicates from one representative biological replicate. Two to three biological replicates were performed. The *LFY* and *MP* probes for in situ hybridization have been described (*Yamaguchi et al., 2013*). *FIL*, and *TMO3* probes were amplified and cloned into pGEM-T (Promega, Fitchburg WI, United States). RNA in situ hybridization was performed as previously described (*Wu and Wagner, 2012*). Inflorescences were harvested and fixed at 24 DAG, before manifestation of the pin inflorescence phenotypes in *mp-S319* and *brm-3 syd-5*. Sections to be directly compared were processed together on the same slide. Protoplasts were transfected as described (*Yoo et al., 2007*). After transfection, or after auxin or mock treatment, protoplasts were harvested; RNA was extracted using the RNeasy Micro kit (Qiagen). cDNA was synthesized from 100 ng total RNA using the superscript III kit (Thermo Fisher). For gene expression analysis in pER>>MPΔC or *ap1 cal* axr3-GR, 4 µg of total RNA was used for reverse transcription with the superscript IV kit (Thermo Fisher).

## ChIP

ChIP was performed as previously described (*Yamaguchi et al., 2014*). The following antibodies were used: anti-GFP (A6455, Thermo Fisher), anti-HA (12CA5, Roche, Basel, Switzerland), anti-Histone H3K9ac antibody (39138, Active Motif, Carlsbad CA, United States) and anti-SYD (*Wagner and Meyerowitz, 2002*). Two to three biological replicates were performed for each ChIP experiment. The *Ta3* retrotransposon (At1g37110) was used as the negative control (NC) locus for all ChIP experiments. Nontransgenic plants of the same age served as ChIP control genotypes. When comparing binding in different genotypes (wild type vs mutant), percent input enrichment in each ChIP sample was normalized over that at the NC locus to compute fold enrichment. To enrich for incipient flower primordia, ChIP experiments displayed in *Figure 2* and *Figure 2—figure supplement 2* were

performed in the *ap1-1 cal-1* (*Ferrandiz et al., 2000*) genetic background. For primer sequences see *Supplemental file 1*.

## FAIRE

FAIRE was performed as described (*Omidbakhshfard et al., 2014*). For inflorescences, 0.3 g of tissue was crosslinked with 1% formaldehyde in crosslinking buffer under vacuum for 8 min. For tests in plant cells, $1 \times 10^6$ protoplasts were crosslinked in 1% formaldehyde, 1x PBS for 8 min. Isolated DNA fragments were further purified by Qiaquick DNA purification columns (Qiagen, Germantown MD, United States). The *Ta3* retrotransposon (At1g37110) (*Johnson et al., 2002*) was used as the NC locus for all FAIRE experiments. qPCR was performed for crosslinked and noncrosslinked FAIRE samples. Fold enrichment was obtained by normalizing DNA accessibility in FAIRE samples over that of un-crosslinked DNA. The fold enrichment at each experimental locus was normalized over that of *Ta3*.

## Protein interaction

To test for interaction between BRM and MP in yeast, full-length MP (amino acids 1–902), MPN (amino acids 1–348), MPM (amino acids 349–765) and MPC (amino acids 766–902) (*Tiwari et al., 2003*) were cloned into pDEST22 (Thermo Fisher). The N-terminal protein interaction domain of BRM (amino acids 1–976) (*Wu et al., 2012*; *Efroni et al., 2013*) was used as bait. The pDEST22 MP constructs and pDEST32 BRM were co-transformed into yeast strain AH109 (Clontech, Mountainview CA, United States). For yeast-three-hybrid analyses, BRM (amino acids 1–976) was fused to the GAL4 DNA binding domain in pBridge (Clontech). Full-length AXR3 or BDL were cloned behind the *MET25* promoter into the same vector. Full-length MP and MPΔC (amino acids 1–765) were cloned into pACT2 (Clontech). Constructs in pBridge and pACT2 were cotransformed into yeast strain AH109. Serial dilutions of transformed cells grown for 72 hr on -Trp-Leu (-Met) and on -Trp-Leu (-Met)-His/SD medium with 0.5 to 0.1 mM 3-amino-1,2,4-triazole (Y2H and Y3H, respectively).

For bimolecular fluorescence complementation (BiFC), the above mentioned fragments of BRM and MP and amino acids 1 to 657 of SYD (*Wu et al., 2012*), were shuffled into pSPYNE(R)173 and pSPYCE(MR) (*Waadt et al., 2008*). 4xMyc-axr3 and 4xMyc-bdl in pUC19 were used for BiFC competition assays. BiFC in protoplasts was performed as previously described (*Yoo et al., 2007*). For each experiment, YFP signal was compared only within protoplast populations prepared and transformed at the same time. Images were taken with a confocal microscope with the same gain (Leica, LCS SL). Multiple images were taken for each biological replicate. The interaction frequency was calculated by counting the number of YFP positive nuclei among all protoplasts under an epifluorescence microscope (Olympus, MVX100). At least one hundred and fifty protoplast cells were counted for each sample; three biological replicate samples were performed for each combination tested.

For co-immunoprecipitation assays, FLAG-BRM plus 3xHA-MP/MPN, FLAG-SYD plus 3xHA-MP/ MPN or FLAG-BRM plus 3xHA-MP/MPΔC plus 4xMyc-AXR3/BDL/PI cloned into pUC19 were co-transfected into *Arabidopsis* leaf protoplasts. PI (PISTILLATA) served as NC protein in the competitions because of its similar molecular mass to AXR3 and BDL. The nuclear fraction of the protoplasts was prepared and co-immunoprecipitation was conducted essentially as previously described (*Ryu et al., 2007*). Anti-FLAG (1:2000; 9A3, Cell Signaling, Danvers MA, United States) was used for immunoprecipitation. Anti-HA-peroxidase high affinity (1:1000; 3F10, Roche), anti-c-Myc (1:2000; C3956, Sigma), or anti-H3 (1:5000, ab1791, AbCam, Cambridge MA, United States) were used for Western blotting. Band signal intensity was quantified using image J (*Schneider et al., 2012*). The signal intensity of immunoprecipitated HA-MP was normalized over that of HA-MP in the input for each sample to obtain percent input enrichment.

For in situ proximity ligation assays (PLA), inflorescences (3 cm bolt) were fixed in 4% paraformaldehyde, $1 \times$ PBS, 0.1% Triton X-100 overnight at 4°C. Inflorescences were dehydrated, embedded and sectioned as for in situ hybridization (*Wu et al., 2012*). The antigen was unmasked by heat-induced antigen retrieval in 10 mM Tris–HCl and 1 mM EDTA (pH 9) for 40 min. Rabbit anti-GFP (1:1600; 2555, Cell Signaling) and mouse anti-HA (1:1200; 6E2, 2367, Cell Signaling) antibodies were applied to sections and incubated overnight at 4°C. PLA was performed according to manufacturer's instructions (Duolink, Sigma) with the following modifications: sections were incubated with PLUS and MINUS PLA probes overnight at 4°C, ligation was performed at 37°C for 2 hr and amplification

was performed at 37°C for 3 hr. Rolling-circle products were visualized with horseradish peroxidase (HRP)-labeled probes (Duolink in situ Detection Reagents Brightfield, Sigma). The number of rolling circle products was counted under a brightfield microscope (Olympus, BX51).

## Micrococcal nuclease (MNase) digestion

2 g of above ground tissue was harvested without crosslinking and nuclei and chromatin were isolated as previously described (*Chodavarapu et al., 2010*) with minor changes. The nuclear pellet was washed twice with HBB buffer. The isolated chromatin was digested with a final concentration of 0.2–0.5 units/µl MNase (Takara, Tokyo, Japan) for 3 min in digestion buffer at 37°C. Subsequent steps were performed as previously described (*Chodavarapu et al., 2010*). Relative nucleosome occupancy was analyzed by tiled oligo qPCR. Percent input enrichment for each primer pair was extrapolated using a dilution series of undigested genomic DNA (*Gévry et al., 2009*). Fold enrichment of nucleosome bound DNA was calculated by normalizing percent input of each primer pair over that of the *gypsy*-like retrotransposon (At4g07700). MNase in protoplasts was performed as in intact tissues with some modification. $2 \times 10^6$ cells were harvest by centrifugation at 11,800 rpm for 2 min, followed by resuspension in 500 µl lysis buffer by vortexing. After centrifugation at 7300 rpm for 5 min, the nuclear fraction was resuspended in HBC buffer (*Chodavarapu et al., 2010*). The chromatin was digested with a final concentration of 0.02 units/µl MNase (Takara). For primer sequences see *Supplemental file 1*.

## Data analysis and presentation

Mean ± SEM is shown for all numerical values, for frequencies the error bars are proportional to the standard error of the pooled percentage computed using binomial distribution $\sqrt{\frac{p(1-p)}{n}}$. For qRT-PCR and ChIP one representative of three experiments is shown. For all other data normal distribution was tested by the Kolmogorov–Smirnov test. For normally distributed data, statistical significance was computed using a two-tailed Student's *t*-test. For non-normally distributed data, statistical significance was computed using a two-tailed Mann–Whitney *U* test. Significance cutoff (*) p < 0.01. NS = Not significant. **p < 0.001, ***p < 0.001. Box and whisker plots: lower vertical bar: sample minimum. Lower box: lower quartile. Red line: median. Upper box: upper quartile. Upper vertical bar: sample maximum. For flower initiation tests, the parental line with the fewest flowers served as control.

## Acknowledgements

We thank J Alonso, M Bayer, JD Wagner, D Weijers and Z Spiegelman for comments, J Kim for advice on statistical analyses and D Weijers and J Reed for materials (*mp-B4149* seed and axr3 clone, respectively). This work was supported by NSF IOS grant 1257111 and MCB-1243757 to DW, by a Japan Society for the Promotion of Science Postdoctoral Fellowship for Research Abroad to NY and by Howard Hughes Medical Institute, Gordon and Betty Moore Foundation, and the NIH (GM43644) to M.E.

## Additional information

### Funding

| Funder | Grant reference number | Author |
| --- | --- | --- |
| National Science Foundation (NSF) | IOS grant 1257111 | Doris Wagner |
| National Science Foundation | MCB-1243757 | Doris Wagner |
| Japan Society for the Promotion of Science (JSPS) | Postdoctoral Fellowship for Research Abroad | Nobutoshi Yamaguchi |
| Howard Hughes Medical Institute (HHMI) | | Mark Estelle |
| Gordon and Betty Moore Foundation | | Mark Estelle |

| National Institutes of Health (NIH) | GM43644 | Mark Estelle |

The funders had no role in study design, data collection and interpretation, or the decision to submit the work for publication.

### Author contributions

M-FW, NY, JX, Conception and design, Acquisition of data, Analysis and interpretation of data, Drafting or revising the article; BB, ME, Drafting or revising the article, Contributed unpublished essential data or reagents; YS, Initiated study, Acquisition of data; DW, Conception and design, Analysis and interpretation of data, Drafting or revising the article

## Additional files

### Supplementary files

• Supplementary file 1. Primers used in this study.

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
