## [Decision Letter]

Thank you for submitting your work entitled “Auxin-regulated chromatin switch directs acquisition of flower primordium founder fate” for peer review at *eLife*. Your submission has been favorably evaluated by Detlef Weigel (Senior Editor) and three reviewers, one of whom is a member of our Board of Reviewing Editors.

The reviewers have discussed the reviews with one another and the Reviewing editor has drafted this decision to help you prepare a revised submission.

All three reviewers thought this was an interesting manuscript. Its demonstration of how SWI/SNF ATPase activity may act as a “gatekeeper” to auxin responses by direct interaction between auxin response factor MP and nucleosome remodeling factors BRM and SYD provides an important mechanistic insight into auxin activity. The results are novel, of broad interest and mostly of high quality, and the reviewers thought the tethering and PLA approaches were elegant. Several experiments need to be improved, or explained more clearly, however, and the reviewers felt some conclusions were not fully justified by the provided data. Details of these issues are listed below and would need to be addressed before the work is suitable for publication in *eLife*.

Essential revisions:

1) There are a number of cases where the phenotypes/gene expression changes attributed to altering MP and SYD/BRM could be due to very indirect effects. For example, finding that genes associated with flower primordium formation are underexpressed in *mp* and *brm/syd* mutants hardly is evidence that these genes are commonly regulated targets of MP and SYD/BRM since the basic, auxin dependent, pre-pattern of the SAM including incipient flower primordia is severely disturbed in these mutants. The authors demonstrate convincingly that MP, BRM and SYD bind to the promoters of *FIL, TMO3* and *LFY*, but that could be coincidence, since TFs bind to thousands of target sites in the genome without noticeable regulatory impact.

Independent evidence for the consequence of this binding would be desirable, e.g., conditional loss-of-function or inducible gain-of-function, which would help to better separate direct from indirect effects.

2) The most interesting and surprising results center around tethering BRM/SYD to MP via BUSHY. This rescue suggests that organ formation can happen independently of auxin signaling, since the fusion can no longer interact with IAAs and thus should be constitutively active. Because this is such an important finding, some further experiments and discussion are needed. First, the MPN-alone control for the phenotype rescue experiments should be reported along with the fused MPN-BSH results. What degree of *mp* rescue is conferred by MPN alone?

The rationale for concluding that these results demonstrate that SYD/BRM are required for auxin dependent development needs to be more fully developed as well. If this is auxin-independent organ initiation and phyllotaxis, this would upset much of the current thought in the field. This is a substantial finding that needs to be placed in context.

3) The paper could use some documentation of the tissue expression pattern of BRM and SYD since interaction with MP is such a strong theme. Although the authors cite the [4] in the discussion for dividing cells, it is not clear how this related to the tissues of concern here. If BRM and SYD expression is published elsewhere, citing this previous work is fine, otherwise, provide it here. Given the nature of BRM and SYD activity, expression is important in addition to location of phenotypic effects.

4) You claim that MP recruits BRM/SYD to chromatin based on the finding that they bind to the same promoter regions using ChIP-PCR. There are two issues: A) from the data presented we cannot judge the resolution of the analysis because amplicons between the enriched regions are missing. B) It has been shown that many TFs bind to the same very small promoter regions (HOT regions) without necessarily binding to the same element or being recruited by one-another (Heyndrickx et al.). Thus, in the absence of high resolution ChIP-seq data, this conclusion of general BRM/SYD recruitment by MP cannot be drawn.

It is essential that the resolution of the BRM/SYD and MP binding sites on shared target promoters is refined by additional ChIP-PCR (or ChIP-seq) mapping along the chromatin in these regions.

There is also an issue of wording here. For example the phrase “occupy the MP-bound sites” sounds like there is evidence that simultaneous binding was demonstrated; however, this would require a reChIP experiment, and such work was not done.

We also note that the finding that BRM and SYD binding to FIL and LFY promoters is lost in *mp* mutants is difficult to interpret due to the dramatic *mp* phenotypes and the resulting cellular reprogramming effects.

5) In classic genetics terms, the additive phenotypes of *brm/syd* mutans with *mp* would suggest that they act independently rather than in a complex. Since the alleles used in this study are poorly described in the text it is hard to follow the reasoning of the authors. For example, what's the evidence that the conditional BRM allele is really a null?

6) Evidence that BRM/SYD can turn a gradient into a binary switch (presented in the Discussion) is lacking, or at least the evidence the authors are drawing upon needs to be pointed out more clearly. Also the conclusion that “BRM or SYD are necessary for the auxin-dependent increase in accessibility at MP target loci in the context of chromatin, a prerequisite for induction of MP targets” has only been shown indirectly.

7) The statistical tests used are adequately described in figure legends but they should describe what error bars represent, e.g. they present both count data and percentage data with error bars so the bars should represent different measures of variance. Just a very short addition to the figure legends is needed right after the statistical test described.

---

## [Author Response]

Essential revisions:

*1) There are a number of cases where the phenotypes/gene expression changes attributed to altering MP and SYD/BRM could be due to very indirect effects. For example, finding that genes associated with flower primordium formation are underexpressed in* mp *and* brm/syd *mutants hardly is evidence that these genes are commonly regulated targets of MP and SYD/BRM since the basic, auxin dependent, pre-pattern of the SAM including incipient flower primordia is severely disturbed in these mutants. The authors demonstrate convincingly that MP, BRM and SYD bind to the promoters of* FIL, TMO3 *and* LFY*, but that could be coincidence, since TFs bind to thousands of target sites in the genome without noticeable regulatory impact*.

*Independent evidence for the consequence of this binding would be desirable, e.g., conditional loss-of-function or inducible gain-of-function, which would help to better separate direct from indirect effects*.

To address this concern, we have tested gene expression changes after inducible gain of MP activity (two-component estradiol inducible MP∆C) and inducible loss of MP activity (dexamethasone inducible *axr3-GR*). Steroid treatment yielded an increase and decrease, respectively, in the expression of *FIL, TMO3, LFY* and *ANT* (Figure 2—figure supplement 2) in agreement with our previous data (70). The combined analyses strongly suggest that all four genes are direct MP regulated targets.

We do not currently have inducible gain or loss of function lines for BRM or SYD. However, not only is expression of MP-regulated target genes reduced in *brm syd* mutants, but it is also increased (*FIL*) upon BRM or SYD complex tethering (via MPN-BSH). Finally, we performed further ChIP experiments as suggested by the reviewers (see also point 4 below). The resulting data (Figure 2—figure supplement 4) show a strikingly similar binding pattern of MP, BRM and SYD at the shared target loci, which is unlikely to be coincidental.

Thus our combined data make a compelling case for direct rather than indirect roles of MP, BRM and SYD in the expression of shared target genes, three of which (*LFY, ANT* and *FIL*) play a role in flower primordium initiation ([70], this study).

*2) The most interesting and surprising results center around tethering BRM/SYD to MP via BUSHY. This rescue suggests that organ formation can happen independently of auxin signaling, since the fusion can no longer interact with IAAs and thus should be constitutively active. Because this is such an important finding, some further experiments and discussion are needed. First, the MPN-alone control for the phenotype rescue experiments should be reported along with the fused MPN-BSH results. What degree of* mp *rescue is conferred by MPN alone?*

*The rationale for concluding that these results demonstrate that SYD/BRM are required for auxin dependent development needs to be more fully developed as well. If this is auxin-independent organ initiation and phyllotaxis, this would upset much of the current thought in the field. This is a substantial finding that needs to be placed in context*.

We do not think that our findings suggest auxin independent organ initiation or phyllotaxis. We agree that one possible outcome of the tethering experiment could have been ‘ectopic’ organogenesis, such as a ring of flower primordia initiating from the peripheral zone of the shoot apex. What we instead observe is rescue of the flower initiation defect of *mp-S319* with flowers initiating essentially in a spiral phyllotaxis. We have not measured the angle at which primordia initiate, so we can only rule out a gross defect in phyllotaxis in *mp-S319* MPN-BSH.

The lack of a dramatic gain-of-function phenotype in the MPN-BSH *mp-S319* plants may be attributable to the residual MP activity present in the hypomorph *mp-S319* mutant, which may interpret the auxin gradient. In this case, the combined activities of MPN-BSH and of the residual MP would direct flower initiation. Alternatively, other factors (other ARFs for example) may contribute to the interpretation of the auxin gradient together with MP (and BRM/SYD) during flower primordium initiation. While further experiments are needed to distinguish between these possibilities, we currently favor the latter hypothesis (MP lacking the Aux/IAA interaction domain is not sufficient for flower initiation) because introduction of MP∆C into *mp* null mutants also did not result in formation of a ring of flower primordia ([34], New Phytologist 2012). We have added a statement to the relevant section of the text to clarify this point.

Finally, as suggested by the reviewers, we have quantified flower number in MPN *mp-S319* and MPNm1 *mp-S319* plants in the same fashion as we did for MPN-BSH (and related constructs). We scored flower number in all *mp* homozygous T1 plants obtained (n>15) and found that neither MPN nor MPNm1 rescued the *mp-S319* phenotype (Figure 4—figure supplement 2 and D). These findings suggest that rescue of the flower initiation defect of *mp-S319* by MPN-BSH depends on its ability to recruit BRM/SYD. Figure 9 shows that the abundance of the endogenous MP mRNA was unaltered in *mp-S319* MPN or *mp-S319* MPNm1 relative to *mp-S319*.

Author response image 1.*MP* message abundance relative to that of *EIF4A*.*MP* abundance is similar in *mp-S319* and *mp-S319* transformed with MPN or MPNm1.**DOI:**
http://dx.doi.org/10.7554/eLife.09269.025

*3) The paper could use some documentation of the tissue expression pattern of BRM and SYD since interaction with MP is such a strong theme. Although the authors cite the*[4]*in the discussion for dividing cells, it is not clear how this related to the tissues of concern here. If BRM and SYD expression is published elsewhere, citing this previous work is fine, otherwise, provide it here. Given the nature of BRM and SYD activity, expression is important in addition to location of phenotypic effects. .*

We should have mentioned prior data that indicating that both BRM and SYD are expressed in incipient flower primordia (63; 67). We have added a sentence to the Results section to remedy this oversight.

*4) You claim that MP recruits BRM/SYD to chromatin based on the finding that they bind to the same promoter regions using ChIP-PCR. There are two issues: A) from the data presented we cannot judge the resolution of the analysis because amplicons between the enriched regions are missing. B) It has been shown that many TFs bind to the same very small promoter regions (HOT regions) without necessarily binding to the same element or being recruited by one-another (Heyndrickx et al.). Thus, in the absence of high resolution ChIP-seq data, this conclusion of general BRM/SYD recruitment by MP cannot be drawn*.

*It is essential that the resolution of the BRM/SYD and MP binding sites on shared target promoters is refined by additional ChIP-PCR (or ChIP-seq) mapping along the chromatin in these regions*.

*There is also an issue of wording here. For example the phrase “occupy the MP-bound sites” sounds like there is evidence that simultaneous binding was demonstrated; however, this would require a reChIP experiment, and such work was not done*.

We agree with these concerns and have performed new ChIP analyses with the previously employed and with additional amplicons that span the upstream intergenic regions of the *FIL, TMO3* and *LFY* loci (Figure 2—figure supplement 4). The overall pattern of BRM, SYD and MP association at each locus is very similar. We also have changed the wording in the text to avoid the impression that we have evidence for simultaneous binding of MP and BRM/SYD.

*We also note that the finding that BRM and SYD binding to FIL and LFY promoters is lost in* mp *mutants is difficult to interpret due to the dramatic* mp *phenotypes and the resulting cellular reprogramming effects*.

We fully agree with this point and have changed the conclusion sentence in the Results section for Figure 2 to state that the data are ‘consistent with the hypothesis that MP may recruit BRM/SYD to target loci’. However, in Figure 7 we show that *axr3-GR* induction leads to BRM and SYD eviction from MP target loci in wild-type morphology plants. Because Aux/IAA proteins can only effectively compete away BRM and SYD when they are complexed with MP (Figure 7), these data provide additional support for a role for MP in recruiting BRM/SYD.

*5) In classic genetics terms, the additive phenotypes of* brm/syd *mutans with* mp *would suggest that they act independently rather than in a complex. Since the alleles used in this study are poorly described in the text it is hard to follow the reasoning of the authors. For example, what's the evidence that the conditional BRM allele is really a null?*

This is an important point that requires clarification of the types of alleles used for the genetic experiments. We have highlighted the allele designations for *brm, syd* and *mp* throughout the revised manuscript. *mp-S319* is a hypomorph allele and as such can also be enhanced by mutations that act in the same pathway (or complex). In addition, the *syd-5* mutant or the strong *aMIRBRM* mutant each represent only partial loss of remodeling activity in this pathway, because BRM and SYD act redundantly in flower primordium initiation (Figure 1). Thus, loss-of either BRM or SYD activity, which does not cause a primordium initiation defect, should enhance the flower primordium initiation defect of *mp-S319* to phenocopy the null mutant (no flower primordia initiate). This is indeed what we observe. The null *brm* mutant (*brm-1*) double mutant with *mp-S319,* in addition, phenocopies the seedling lethality of the null *mp-B4149* mutant (Figure 1—figure supplement 1).

We have removed the reference to *aMIRBRM* as a conditional ‘null’ mutant and now refer to it as a strong mutant instead. *aMIRBRM* has much reduced BRM message accumulation in incipient flower primordia on the basis of in situ hybridization (67) but we cannot rule out that it still has residual BRM activity.

*6) Evidence that BRM/SYD can turn a gradient into a binary switch (presented in the Discussion) is lacking, or at least the evidence the authors are drawing upon needs to be pointed out more clearly. Also the conclusion that “BRM or SYD are necessary for the auxin-dependent increase in accessibility at MP target loci in the context of chromatin, a prerequisite for induction of MP targets” has only been shown indirectly*.

We agree with the reviewers/editors on both points and have removed the binary switch sentence from the Discussion. For the second point (conclusion sentence after Figure 5), ‘necessary’ in our opinion denotes a condition that has to be met before an event can occur, this includes both direct or indirect control of the event. We have therefore left this sentence unchanged.

However, the combined expression and accessibility data for *brm syd* mutants and remodeling complex tethering by MPN-BSH strongly point to a direct role of BRM/SYD in the accessibility at MP targets in the context of chromatin and in the transcriptional induction of MP targets (see also response to points 1 and 4 above).

*7) The statistical tests used are adequately described in figure legends but they should describe what error bars represent, e.g. they present both count data and percentage data with error bars so the bars should represent different measures of variance. Just a very short addition to the figure legends is needed right after the statistical test described*.

We have corrected this oversight and added the following sentence to the figure legends for all frequency data (BiFC) as well as to the Methods section. The error bars for the frequency data are proportional to the standard error of the pooled percentage computed using binominal distribution.